# 2D/3D Wound Segmentation and Measurement Based on a Robot-Driven Reconstruction System

**DOI:** 10.3390/s23063298

**Published:** 2023-03-21

**Authors:** Damir Filko, Emmanuel Karlo Nyarko

**Affiliations:** Faculty of Electrical Engineering, Computer Science and Information Technology Osijek, Josip Juraj Strossmayer University of Osijek, HR-31000 Osijek, Croatia

**Keywords:** chronic wound, segmentation, measurement, 2D, 3D, active contour model, convolutional neural network, robot

## Abstract

Chronic wounds, are a worldwide health problem affecting populations and economies as a whole. With the increase in age-related diseases, obesity, and diabetes, the costs of chronic wound healing will further increase. Wound assessment should be fast and accurate in order to reduce possible complications and thus shorten the wound healing process. This paper describes an automatic wound segmentation based on a wound recording system built upon a 7-DoF robot arm with an attached RGB-D camera and high-precision 3D scanner. The developed system represents a novel combination of 2D and 3D segmentation, where the 2D segmentation is based on the MobileNetV2 classifier and the 3D component is based on the active contour model, which works on the 3D mesh to further refine the wound contour. The end output is the 3D model of only the wound surface without the surrounding healthy skin and geometric parameters in the form of perimeter, area, and volume.

## 1. Introduction

Chronic wounds are slow to heal, and if ineffective treatment is used, the healing process may be further delayed. Clinicians need an objective method of wound assessment to determine whether current treatment is appropriate or needs to be adjusted. Measuring wounds accurately is an important task in the management of chronic wounds since changes in the physical parameters of the wound are signs of healing progress.

The analysis of chronic wounds mainly involves contact and non-contact methods. Contact methods, including alginate molds, transparency tracing, manual planimetry with rulers and injection of color dyes, are considered traditional and were the most commonly used in the past [1,2]. These methods are usually impractical for medical personnel and very painful for patients. Since wounds can be of any shape, these methods are also often inaccurate and imprecise. Increasing computational capabilities of modern hardware has boosted the application of non-contact wound analysis. Additionally, progress in data analysis has led to the accelerated increase in the application of digital imaging in wound assessment. Marijanovic et al. [3] provide a recent overview of chronic wound analysis using non-contact methods.

Since the wound might theoretically be located on any part of the body and could be of any size or shape, the wound recording process is frequently challenging. The majority of chronic wounds that are discussed in this paper are typically seen on the back or on the legs. Back wounds, e.g., pressure ulcers, typically occur on flatter surfaces, but are often much greater in size than leg wounds (Figure 1a). On the other hand, leg wounds, such as venous and diabetic ulcers, are typically shallow and located on areas of the body that are highly curved (Figure 1b).

Chronic wounds can have a dynamic surface geometry because they experience expansion and reduction phases during the course of treatment. As a result, some areas of the wound may occlude other areas when viewed from specific angles. The recording technique can be rather challenging when reconstructing 3D models of such wounds, involving numerous phases and recording poses. This can be quite tiring if done manually with a hand-held 3D camera or sensor, and since human operators lack precision, such reconstructed 3D models may, at best, miss some details or, at worst, have anomalies [4].

Recently, an automated system has been developed that has a much higher precision than human operators and is able to record wounds from different viewpoints. It tracks the state of the recording process and enforces a specified density of surface samples on all parts of the recorded wound surface [5]. The research presented in this paper is based on this developed system and extends the idea of a full automated system that outputs precise geometric measurements of wounds. Physicians can monitor patients’ progress and promptly administer the right therapy with the help of such measurements and the tracking of their development over time.

The research presented in this paper is comparable to that in [4], which also focuses on the 3D reconstruction, segmentation, and measurement of chronic wounds, but uses very different technologies. The authors in [4] used handheld RGB-D cameras, which are significantly cheaper, but have significant drawbacks in terms of depth accuracy and the influence of surface features and lighting conditions. Because they were handheld cameras, the accuracy of the reconstruction was also affected by the experience of the operator. In the current research, a sophisticated 7-DoF robotic arm is used with an industrial high-precision 3D scanner attached to the end effector to enable a fully automated and accurate 3D reconstruction process.

In order to facilitate the measurement of physical parameters, a precise segmentation of the wound surface from the reconstructed 3D model needs to be performed, which is the main topic of this paper. A segmented wound would enable the measurement of the perimeter of its border, and in the case of surface wounds, its area. For wounds with greater depth, a virtual skin top must be generated, which then enables the calculation of the area of the virtual skin surface and its enclosed volume.

The main scientific contribution of this paper is a novel segmentation algorithm using a combination of 2D and 3D procedures to correctly segment a 3D wound model. The segmentation of multiple 2D photographs per wound is driven by a deep neural network in the form of the MobileNetV2 classifier, which is then optimally combined with a single 3D model and initialization of the initial wound contour. This initial wound contour on a reconstructed 3D model is then optimized and adjusted by an active contour model, which then tightly envelops the actual wound surface using surface curvature to achieve its objective.

The remainder of the paper is organized as follows. A brief overview of relevant research is provided in Section 2. A hardware and software setup of the created system is described in Section 3. Section 4 describes the implementation of individual components of the segmentation algorithm. Section 5 discusses the performance of the developed algorithm, while Section 6 concludes the paper.

## 2. Related Research

The technique of assigning each pixel of an image into one of two categories, wound and non-wound, or separating the wound area from the rest of the image (surrounding healthy tissue or image background), is known as wound segmentation. The accuracy of segmentation is essential for various wound analysis activities such as tissue categorization, 3D reconstruction, wound measuring, and wound healing evaluation. Extracting the visual features of each location is essential for identifying the wound because the wound area typically has different visual features than the healthy skin.

Researchers have employed a variety of approaches to perform 2D wound segmentation, including using K-means clustering [6,7], deep neural networks [8,9,10,11,12,13,14,15], support vector machines [16,17], k-nearest neighbors [4], and simple feedforward networks [18]. Other approaches include using superpixel region-growing algorithms, color histograms, or combined geometric and visual information of the wound surface to segment wounds.

A systematic review of 115 papers dealing with image-based AI in wound assessment was performed by Anisuzzaman et al. [12]. Their final conclusion was that each of the mentioned approaches had some limitations and, hence, no method could be said to be preferable to the others. The most popular methods by far implement deep neural networks. 

A deep convolutional neural network architecture called MobileNetV2 was proposed by Wang et al. [11] for wound segmentation. The network was pre-trained using the Pascal VOC dataset prior to training. The output of the trained neural network model was a segmented grayscale image of the wound, with each pixel indicating the probability of representing a wound pixel. This image then underwent several post-processed steps: thresholding to initial create a binary image, hole filling, and the removal of small regions, thereby resulting in a final binary image or segmentation mask. In the same paper, the authors proved the superiority of their model by comparing with four other deep neural network models (VGG16, SegNet, U-Net, and Mask-RCNN) using the Medetec dataset [19]. 

A segmentation technique made up of the U-Net and LinkNet deep neural networks was proposed by Mahbod et al. in [13]. These deep neural networks are basically encoder–decoder convolutional networks. These networks were pre-trained using images from the Medetec database [19], and then trained on the MICCAI 2021 Foot Ulcer Segmentation (FUSeg) Challenge dataset [20], thereby resulting in two separate models. Both models evaluate the test image, and the combined output of their evaluations yields the final result.

Scebba et al. [14] implemented an automated approach to wound detection and segmentation using specialized deep neural networks consisting of three steps: a wound detection neural network that detects the wound(s) on the raw wound image; a processing module that performs cropping, zero padding and image resizing to exclude uninformative background pixels; and the final segmentation model that also includes a deep neural network model. The results showed that the fusion of automatic wound detection and segmentation improved segmentation performance and enabled the segmentation model to generalize well to images of wounds that are not in the distribution.

Marijanović et al. [18] proposed a method for wound detection with pixel-level instance segmentation, which consists of an ensemble of three simple feedforward networks, each comprising only five fully connected layers. For each of the feedforward neural network classifiers, input data were created using a conventional fixed-size overlapping sliding window method, with the sliding window sizes varying for each classifier. Post-processing involving thresholding, morphological closure, and morphological opening was performed on each of the predicted outcomes or probability maps of the respective neural network classifiers. The logical operation AND was then used to merge these binary post-processed images obtained as the output predictions of the three neural networks. The ensemble classifier suggested by the authors outperformed Wang et al.’s technique [11] in terms of detection and processing time and proved to be relatively robust to image rotations. Training and testing were conducted using data from the MICCAI 2021 FUSeg Challenge [20].

The segmentation of wounds from 3D surfaces such as meshes is far less popular in the literature since it often requires specialized hardware for acquisition. However, even when regular cameras are used, extension into the third dimension is often cumbersome and requires specific knowledge to analyze and use such data.

In medical and other research, lasers are frequently employed for 3D reconstruction, where a laser line projection sensor calibrated with an RGB camera can produce precise and colored 3D reconstructions. One of the earliest studies to implement such a method was Derma [21], where the Minolta VI910 scanner was employed by the authors. Laser and RGB camera technology was also utilized in related studies [22,23]. These systems have been shown to be extremely accurate, but they are also difficult to operate. Furthermore, these investigations had the limitation that the full wound must be seen in one frame.

In order to improve image-based techniques and provide more accurate measurement, some wound assessment systems use 3D reconstruction. As a result, multiple view geometry algorithms using conventional cameras are frequently employed. In [24], the authors create a 3D mesh model using two wound images collected at various angles. The final 3D mesh has a low resolution as a result of the technology and techniques used.

Some research tries to combine 2D and 3D information to enhance the operation and increase the measurement precision. 

To find the center of the wound, Filko et al. [4] include a 2D detection phase in the 3D reconstruction procedure inspired by Kinectfusion. The kNN method and color histograms are used to implement this. Additionally, they segment the wound from the reconstructed 3D model by first dividing the reconstructed 3D surface into surfels. Then, utilizing geometry and color information to create relationships between neighboring surfels, a region-growing process groups these surfels into larger smooth surfaces. Finally, using spline interpolation, the wound boundary is determined and the wound is then isolated as a distinct 3D model and its perimeter, area, and volume are calculated.

Niri et al. [25] employed U-Net to roughly segment the wound on 2D images and used structure from motion algorithm to reconstruct the 3D wound surface from a sequence of images. They then used reprojections of the 3D model to enhance the wound segmentation on the 2D input images as well as the 3D model. They managed to measure the wound area, but since the ground truth employed is based on the models acquired by the same technique, the accuracy of the actual area measurements is not fully validated.

In a later study, Filko et al. [5] developed a robot-driven system for the acquisition and 3D reconstruction of chronic wounds, which also utilized 2D segmentation based on neural networks as its wound detection subsystem. The research presented in this paper is the continuation of that research.

The majority of research is focused on segmentation on 2D images, especially employing deep learning that has excellent properties proven over the myriad other applications. In this research, deep learning is also employed in order to generate an initial, rough wound segmentation, which, because of the errors in camera calibration and imprecision of the 2D segmentation, requires further adjustment when projected onto the 3D model. The 3D side of the segmentation is based on the application of a 3D active contour model that further refines the original contour by utilizing surface curvature to find more optimal wound borders on the 3D mesh model of the wound and its local surroundings. This novel combination of deep learning 2D segmentation and 3D refinement using an active contour model is the main contribution of this paper.

## 3. Hardware and Software Configuration

The hardware configuration (Figure 2) of the acquisition system consists mainly of a Kinova Gen3 7-DoF robot arm and a Photoneo PhoXi M 3D scanner. The Kinova Gen3 robot arm has an RGB-D camera based on Intel RealSense technology embedded in its tool link in the form of the Kinova vision module. The Photoneo PhoXi 3D scanner is connected to the Kinova Gen3 tool link via a custom 3D printed frame. The Kinova RGB camera was manually calibrated to the PhoXi 3D scanner, while the PhoXi 3D scanner was also manually calibrated to the Kinova Gen3 tool link, which enabled transformations between PhoXi and robot base reference frames.

All experiments in this paper were performed on two Vata Inc. medical models (Figure 3):Seymour II wound care model, which includes stage 1, stage 2, stage 3, and deep stage 4 pressure injuries, as well as a dehisced wound.Vinnie venous insufficiency leg model, which includes various injuries as well as venous ulcers and foot ulcers.

## 4. Wound Segmentation

Wound segmentation is an important step in obtaining physical measurements of the wound such as area, perimeter, and volume. The estimation of these parameters requires the reconstruction of a 3D model of the actual wound. As mentioned earlier, the segmentation algorithm is built upon a robot-driven wound detection and 3D reconstruction system [5]. Therefore, for the sake of completeness, the description of those prior phases will be included in the next subsection.

### 4.1. Wound Detection and 3D Reconstruction

The wound 3D reconstruction system is divided into six main stages (Figure 4):Wound detection;Moving the robot to chosen pose and recording;Point cloud alignment;Point cloud analysis;Hypothesis creation and evaluation;Recording pose estimation.

The first step in the system operation is to detect the wound, which must be located in front of the robot. The purpose of detection is to focus the reconstruction process to a relatively small volume instead of reconstructing the entire scene in front of the robot. During the wound detection process, the system acquires an RGB-D pair of images using the Kinova vision module. The RGB image is used for 2D wound detection by a neural network classifier, while the depth image is used for establishing the position of the wound in 3D space. 

The second stage is to control the robot to the desired recording pose. During this stage, the PhoXi scanner acquires depth images and point clouds, while the RGB image acquired by the Kinova vision module is registered to the PhoXi depth image. In the case that the considered point cloud is the first in a series for the wound reconstruction process, an additional wound detection is executed in order to create a volume-of-interest bounding box, which is then used as the region to concentrate the efforts of the reconstruction process and the segmentation process in the later stages.

The alignment of the acquired point clouds with the ones from previous recording cycles is the objective of the third stage. In the case of the initial recording, the alignment is skipped; if it is the second recording, a pairwise alignment between the previous and current point cloud is performed. In the case of the third and every subsequent recording, a full pose graph optimization is performed using all recorded point clouds up until that point in time.

The fourth stage focuses on analyzing the reconstructed surface by determining the surface deficiencies such as surface density and surface discontinuities by the classification of points included in the volume-of-interest bounding box in four classes: core, outlier, frontier, and edge.

In the fifth stage, a list of hypotheses is generated that are used as the next best view for the surface reconstruction process. A hypothesis list is, in part, populated by hypotheses generated using the surface point density data consisting of clustered poses generated from each frontier point. The other part of the hypothesis list is generated using discontinuity data consisting of structures called DPlanes, which are created by clustered edge points.

The sixth and final stage checks whether the evaluated hypotheses in the list are accessible by the robot. If a hypothesis is accessible, it is then chosen as the next best view for acquisition. If it is not accessible, the system tries a number of adjusted views in the vicinity of the considered hypothesis and tests whether they are accessible instead.

The wound reconstruction stops if no further hypothesis is created or if none of the hypotheses or their adjacent views are accessible. The final reconstructed point cloud is created by voxel filtering of the complete point cloud created by the alignment of the acquired point clouds. From this final point cloud, the points enveloped by the bounding box volume-of-interest are cropped and sent to the next stage of the wound analysis process, which is the segmentation stage. A complete description of the wound reconstruction system can be found in [5].

### 4.2. Wound Segmentation Algorithm

The input for the wound segmentation algorithm consists of a final 3D reconstructed wound model in the form of a 3D point cloud, RGB-D pairs of images, and final (optimized) poses of the recordings used to create the 3D model.

The wound segmentation algorithm includes five stages: Two-dimensional per-pixel wound segmentation of RGB images made by Kinova robot vision system using MobileNetV2 classifier.Registering binary masks created by the previous stage to the depth image created by PhoXi 3D scanner.Optimized labeling of wound 3D model points using registered binary masks.Mesh subdivision to improve mesh density.Active contour model to refine wound segmentation on a 3D model.

#### 4.2.1. Per-Pixel 2D Wound Segmentation

The 2D wound segmentation procedure used in this paper is based almost entirely on the method proposed by Wang et al. [11], with two exceptions. First, our own database of images was used in model training and, secondly, an additional postprocessing procedure, utilizing GrabCut image segmentation [26], was included in the final stages in order to improve the obtained results. The output of the classifier is a binary mask marking the wound area(s).

For the purposes of our research, a database of 145 images of two wound models (the Seymour II Wound Care Model and the Vinnie Venous Insufficiency Leg Model by VATA Inc.) was created. Thus, the classifier model obtained in this work is only suitable for images of synthetic wounds. The original images, of dimension 1280 × 780, were taken under uncontrolled illumination conditions, with various backgrounds. Sample images of the dataset are shown in Figure 3. The images were manually annotated per pixel into wound and non-wound. This dataset was further augmented (image flipping and rotation by 180°) and then divided into a training set with 504 images and a test set with 76 images. 

In order to implement the method proposed by Wang et al. [11], the images were resized, i.e., downscaled to the dimensions of 244 × 244. After segmenting using the MobileNetV2 classifier, the segmented image was resized to its original size (upscaled). Since the MobileNetV2 classifier is a per-pixel classifier, the wound segment on the resized or upscaled segmented image is blocky. In order to refine the results, the GrabCut image segmentation method [26] was used to further improve the segmented image whereby the ROIs obtained as outputs of the MobileNetV2 classifier serve as the initial input to the GrabCut segmentation procedure. 

This is shown with the aid of the images shown in Figure 5 and Figure 6. Figure 5 displays one of the test images (Figure 5a) with a section enlarged (Figure 5b). This enlarged section is further displayed in Figure 6 for different stages of the segmentation procedure.

Figure 6a shows the binary image obtained as the output of the trained MobileNetV2 classifier after resizing from the dimension 244 × 244 to the original image dimension (1280 × 780). By superimposing these pixels onto the original image, the wound pixels on the original image are marked (Figure 6b). It can be noticed that the edges of the wound area are blocky or pixelated. By using the original image as well as the corresponding ROIs marked with bounding boxes in Figure 6c as inputs to the GrabCut image segmentation procedure, the obtained wound areas marked in Figure 3d are visibly improved compared to Figure 6b. Comparing the wound areas in Figure 6b,d, it can be noticed that after the additional postprocessing stage, i.e., GrabCut segmentation, the wound areas and the boundaries of the wounds are better defined.

#### 4.2.2. Registering Binary Masks

Binary images or masks created in the previous stage of 2D segmentation need to be registered with the PhoXi depth images in order to be able to apply them to the reconstructed 3D wound model. Prior to registering the masks, they are first dilated by a 30-pixel dilation filter in order to ensure that the mask covers the whole wound in each of the recordings used. This is carried out due to the imperfect camera calibration procedure which has sub-pixel to sometimes even pixel reprojection error at certain distances, as well as the imperfect segmentation procedure in the 2D segmentation stage. This over-segmentation is optimized by the active contour model in the later stage in order to ensure a tighter fit of the detected wound edge on the actual 3D wound model.

In Figure 7, an example of the registration process can be seen, where Figure 7a shows the input RGB image made by the Kinova vision module. Figure 7b shows the output binary mask, while Figure 7c shows the dilated mask. The registered RGB and mask images are showed in Figure 7d and 7e, respectively.

#### 4.2.3. Optimized Labeling of 3D Points

The final wound point cloud output by the acquisition system [5] is processed by voxel filtering, which averages the point positions, colors, and normals for points contained in a given voxel. The resulting point position on the 3D model is not directly referenced in any of the input point cloud recordings, so each point in the point cloud used as input to the segmentation procedure must be reprojected onto each of the input registered binary masks, and then the reprojection that is best suited for the individual point is selected. Furthermore, the optimized labeling and later stages of the segmentation algorithm are performed only on the local wound area point cloud designated by the bounding box volume-of-interest generated during the detection phase of the reconstruction process [5], thereby removing the remainder of the reconstructed scene that is not needed for the analysis of a particular wound.

The algorithm for optimized labeling includes the following steps:Un-project each point in the point cloud on each of the input registered masks, retrieve its mask value and the measured depth value (*d_m_*) from the associated depth image. Also keep the observed depth value the reprojected point would have on the input registered mask (*d_o_*).Calculate the score for each combination of 3D points and input registered masks in the following way:
(1)$$score=\frac{1}{||P-C_{p}||}\cdot \frac{1}{deg\left(acos\left(-N\cdot C_{RZ}{}^{T}\right)\right)}$$
where *P* is the point coordinates, *N* is the normal vector at point *P*, *C_p_* is the camera position where the image was taken, and *C_RZ_* is the Z column of the camera pose rotational matrix.Choose the optimal source of the binary mask label that has the highest score and minimal difference between the measured depth values (*d_m_*) of the original recorded depth image and the calculated, observed depth (*d_o_*) value for each of the point cloud’s points.

The difference in the depth values (*d_diff_*), as seen in Figure 8, is used to detect occlusion when a 3D point would choose a particular registered mask due to a better conditioned relation between the surface normal and camera recording orientation; however, the measured depth (*d_m_*) at that reprojected pixel shows a different point closer to the camera than the observed depth (*d_o_*), which is calculated by the reprojection of the 3D point. Figure 8 distinguishes two camera positions designated as 1 and 2, where position 1 has a better conditioned angle between the recording orientation and surface normal, but has a disadvantaged difference between the observable and measured depth. Position 2 is the opposite of position 1 regarding favorability, but since it does not have penalties regarding depth difference, it will be chosen as the optimal position even though its recording is not in a very good position to record that particular point on the surface.

Figure 9 shows an example of optimized labeling where four recordings were used to reconstruct a wound. The figure shows a reconstructed 3D model, registered RGB, and mask images, as well as local point cloud wound area textured with color and an optimized mask projection.

After the point cloud has been labeled by optimal reprojection, an initial mesh is created using the greedy point triangulation algorithm (GPT) [27] and the initial wound contour is designated by finding mesh vertices that have at least one neighbor labeled as non-wound. That contour is further subsampled by using only half of the points to create an initial contour for the active contour model used in the next phase. Figure 10 shows an example of the initial wound contour on a meshed local wound area.

#### 4.2.4. Mesh Subdivision

Mesh subdivision, in general, is an algorithm that takes a course mesh as input and produces a more dense mesh by subdividing mesh cells into additional cells. This subdivision typically produces an approximated version of the original surface geometry. There are several popular algorithms such as Loop [28], Butterfly [29], or Midpoint [30] for subdividing triangle meshes. Loop and Butterfly both produce approximate surfaces by interpolating curves, while Midpoint preserves the original mesh geometry. To avoid having to make additional assumptions about the scanned wound surface, the Midpoint algorithm is used in this research. The Midpoint algorithm, in each iteration, basically cuts every mesh edge in half and generates four new triangles out of each original triangle. Figure 11 shows an original mesh and the mesh subdivided by the Midpoint algorithm.

Refining the reconstructed wound mesh by increasing the density of triangles and vertices greatly improves the performance of the active contour model (ACM) algorithm, explained in the next subsection, by giving each contour node more freedom to choose a more suitable point that minimizes the energy term (2). The original wound mesh made by the GPT algorithm can be too restrictive for the ACM algorithm even though the original surface is sampled at the millimeter scale, especially in the case of wounds of small size. Figure 12 shows the change in ACM performance with the same configuration when using original wound mesh or subdivided mesh. In this research, two iterations of the Midpoint algorithm were applied for the input wound meshes.

#### 4.2.5. 3D Active Contour Model

The active contour model (ACM) [31] is an algorithm that enables users to find the contours of arbitrary objects in primarily 2D images. ACM is basically a deformable spline influenced by some predefined forces. These forces typically include the attraction force between the nodes of the contour, which causes the contour to contract (or repulse in the case of an expanding contour), and a smoothing force, which counteracts the deformation of the contour. Besides these forces, in order for the ACM to work, the nodes of the contour must be attracted toward a boundary that the user is trying to find—in 2D images, this is typically some kind of gradient, for example, finding the edges in an image with the Sobel filter and then blurring it with the Gaussian filter to have a wider attraction range.

The basic energy functions for the 3D adaptation of the ACM are similar to the generally known 2D case [31]: (2)$$E_{total}=E_{mesh}+E_{contour}$$
(3)$$E_{mesh}=-\sum_{i=0}^{n-i}M\left(i\right),\;M\left(p\right)=max\left(eig\left(C_{p}\right)\right)$$
(4)$$E_{contour}=\alpha E_{elastic}+\beta E_{smooth}$$
(5)$$E_{elastic}=\sum_{i=0}^{n-i}\sum_{j=0}^{k-1}\left|{\left(x_{ki+1}-x_{k}\right)}^{2}+{\left(y_{ki+1}-y_{k}\right)}^{2}+{\left(z_{ki+1}-z_{k}\right)}^{2}\right|$$
(6)$$E_{smooth}=\sum_{i=0}^{n-1}\left|{\left(x_{i+1}-2x_{i}+x_{i-1}\right)}^{2}+{\left(y_{i+1}-2y_{i}+y_{i-1}\right)}^{2}+{\left(z_{i+1}-2z_{i}+z_{i-1}\right)}^{2}\right|$$
where *E_total_* is the cumulative contour energy calculated over all contour nodes that needs to be minimized. It comprises mesh energy *E_mesh_* and contour energy *E_contour_*. Mesh energy, in this case, is the curvature calculated using principle component analysis (PCA) over a list of normals for points in the vicinity of a particular mesh vertex. Basically, it is the largest eigenvalue of the covariance matrix *C_p_* calculated for the list of normals for a particular point *p*. Choosing the correct neighborhood size radius for calculating the PCA is crucial for the attraction force and reach of the ACM, as can be seen in Figure 13b,c, where two different neighborhood radii were used for the calculation. In this research, a neighborhood radius of 5 mm was used for calculating the PCA. The contour energy is further composed of the elastic energy *E_elastic_*, which regulates contraction (or expansion), and smoothing energy *E_smooth_* that regulates the deformability of the contour. The symbols *α* and *β* control the influence of elastic and smoothing energies in the overall energy term. In this research, *α* and *β* were used with the value of 1. The elastic energy is determined by calculating the Euclidean distance between the neighboring nodes of the contour. Since we have a mesh in the 3D case, the distance is composed of the Euclidean distances of all neighboring vertices along the shortest path between contour nodes. Therefore, the elastic energy is basically the geodesic distance along the surface of the triangle mesh between two nodes of the contour. The smoothing energy is calculated as the 3D gradient between the nodes of the contour.

As established earlier, the 3D data of the local wound region that enters this stage of the algorithm is a 3D mesh. In order to better adapt the ACM model to the 3D data, the mesh is used to create a weighted graph. The graph comprises nodes as mesh vertices, the graph edges are triangle edges, and the weights are Euclidean distances between vertices. The 3D ACM algorithm works iteratively in five steps:Determine the current neighborhood for each contour node on the graph.Calculate the new position for each node in a greedy manner.Estimate a spline based on the new node positions.Uniformly sample the spline with the same number of nodes as the initial contour.For each spline sample, find the nearest node on the graph (mesh).

After the initial contour generation in the previous stage of the segmentation algorithm, each list of contour nodes contains the same number of nodes in the following iterations. At the start of each iteration, each contour node generates a list of neighbors that are located a maximum number of graph nodes from the contour node being considered. For this research a two-node neighborhood was found to be the optimal solution. Figure 13d shows a contour, nodes, and neighbors for each of the displayed contour nodes in a two-node-wide range. 

The new position for each node in each iteration is considered in a greedy manner, i.e., independently of the new positions for neighboring contour nodes. The new position for each node is selected from a list of neighbors that minimizes the energy term (2).

Following the designation of the new graph position for every node, a spline is estimated through these positions, which is then sampled in the same number of points as the original list of contour nodes. Since these samples may or may not be located on the wound mesh, a nearest mesh vertex is located by using kd-tree. Figure 13e shows the initial contour as well as a 10-iteration ACM optimized one for the synthetic example of the hole (Figure 13a), where the ACM successfully found the requested hole boundary. In this research, 10 iterations of the ACM were utilized to optimize the initial contour in each of the experimental wounds.

A more realistic example of the ACM application can be seen in Figure 14. Figure 14a–c shows a very successful application of the ACM where the first image shows the masked mesh and initial contour and the second and third images show the initial and final contour with a curvature texture and RGB texture, respectively. A successful run with some inconsistencies can be seen in Figure 14d–f, where the ACM managed, for the most part, to find the wound border with some small areas in the left and top still being over-segmented. The reason for the error on top of the wound was actually the close proximity of the second wound, not object of this analysis, on the same medical model that has more pronounced edges which then “stole” the contour from the observed wound. An unsuccessful run is shown in Figure 14g–i, where it can be seen that the ACM under-segmented the wound, with the final contour slipping to the bottom border of the wound surface instead of remaining on the top border. The error was caused by a relatively shallow wound with strong top and bottom borders. The ACM could not numerically distinguish between the top and bottom border since the bottom was very close due to the shallowness; it therefore “slipped” to the bottom, causing the wound to be under-segmented.

## 5. Results

Generating the final wound boundary using the ACM algorithm is the starting point of calculating the physical properties of the wound, as the boundary directly enables the calculation of the wound perimeter and the area in the case of a shallow wound. For deep wounds, the area is considered to be the skin missing on the top of the wound, therefore a virtual skin top needs to be generated. Similar to our previous research [4], the top of the wound, as well as other holes in the 3D model, are closed using constrained 2D Delaunay triangulation implemented in the VTK library [32]. Even though this Delaunay implementation is for 2D point sets, 3D data can be used by projecting all of the points on a plane chosen as a most likelihood plane by calculating the PCA and choosing the eigenvector corresponding to the largest eigenvalue. Creating a virtual skin top facilitates wound area measurement while creating covers for all holes; it enables the creation of the watertight 3D model and calculating the volume. Figure 15 shows an example of the final wound surface cut from the input mesh by the contour generated from the ACM, along with the generated surfaces used for hole filling.

### 5.1. Case Study

The case studies considered here are wounds from two realistic and lifelike medical models by Vata Inc. as described in Section 3. The Seymour II wound care model also has ground truth (GT) measurements available as mentioned in [4], and the GT measurements can be seen in Table 1.

In this section, we will consider various wound geometries, namely, Stage 3 and 4 pressure ulcers as well as dehisced surgical wound from the Seymour II wound care model. From the Vinnie venous insufficiency leg model, we will consider two venous ulcers as well as a neuropathic ulcer.

#### 5.1.1. Stage 3 Pressure Ulcer

This is a stage 3 pressure ulcer with moderate depth and tunneling on two separate sides. While the used robot-driven 3D wound reconstruction system [5] could be very precise, it was struggling to find a reachable pose to scan the tunneling parts of the wound, therefore those parts of the wound remained largely unscanned in both pressure ulcers considered here. The wound reconstruction can be seen in Figure 16a along with the segmentation performance of Figure 16b–d. The wound was reconstructed from four recordings made from separated viewpoints. As can be seen in Figure 16b, the optimized labeling of the reconstructed mesh was quite successful, with very little unwanted labeling in the surrounding mesh around the wound, which resulted in a tight initial contour prior to ACM. The ACM further optimized the contour, as can be seen from Figure 16c as a green contour. This figure also shows the curvature intensities on the surface of the model. The segmented and cut wound mesh can be seen in Figure 16d. All of the covers for this particular wound can be seen in Figure 15, while the covers for other wounds that have them are not shown because they are quite similar.

The wound perimeter was measured to be 171.09 mm, the area was 2302.0 mm^2^, and the volume was 22,532.09 mm^3^. When comparing the results to GT, they produce a percentage error of 2.62% for perimeter, 2.86% for area, and 4.59% for volume.

#### 5.1.2. 5½ Inch Long Dehisced Surgical Wound

This wound is a 5 ½ inch long dehisced surgical wound with considerable depth and high-angled sides. Due to its high-angled sides (compared to the camera projection plane), it makes reconstruction of this wound a challenge since the projected laser pattern by the 3D scanner does not reflect enough toward the scanner in order to be visible from many view angles reachable by the robot recording system. Therefore, even though the wound is rather simple and could be reconstructed if placed in an unrealistic position for the patient, since it simulates a patient wound on a model of a part of the human body, it could only be partially reconstructed with one side fully scanned, taking into consideration all of the imposed limitations. The reconstructed 3D model can be seen in Figure 17a: the reconstruction was made from four recordings. Figure 17b shows the optimized mask of the detected wound and initial contour, while Figure 17c shows the initial and ACM contour on the surface textured with curvature. Figure 17d shows the segmented and cut wound.

The wound perimeter was measured to be 275.22 mm, the area 2549.63 mm^2^, and the volume 29,764.54 mm^3^. The results, when compared to GT, give a percentage error of 0.89% for the perimeter, 6.23% for the area, and 5.95% for the volume.

#### 5.1.3. Stage 4 Pressure Ulcer

This wound is the second pressure ulcer and the largest and most complex wound on the Seymour II wound care model since it has a very large area, great depth, and tunneling in two different directions. Similarly, as in the previously described pressure ulcer, the tunneling parts of this wound were not able to be reconstructed. Figure 18a shows the reconstructed surface, while Figure 18b shows the optimized mask labeling made from seven recordings from which the wound was originally reconstructed. Figure 18c shows the initial and ACM contour on a model textured with curvature values, while Figure 18d shows the segmented and cut wound. As can be seen in the figures, the ACM failed in 10 iterations to tightly envelop the wound from the bottom and left parts, resulting in a higher area error percentage. Increasing the number of iterations to more than 10 might have helped a little, but reducing the value of β would probably have helped more as it would have made the contour more deformable.

The wound perimeter was measured to be 351.53 mm, the area 7689.11 mm^2^, and the volume 109,839.24 mm^3^. The results, when compared to GT, give a percentage error of 4.96% for the perimeter, 8.02% for the area, and 8.94% for the volume.

#### 5.1.4. Neuropathic Ulcer

This wound is a neuropathic ulcer of comparatively smaller size compared with the previously considered wounds, but because of its depth, four recordings were needed to reconstruct it properly. Figure 19a shows the 3D reconstruction model of the local wound, while Figure 19b shows the optimized mask projection. It can be seen that the mask encompasses a large area around the actual wound. Figure 19c shows that despite this larger masked area, the ACM manages to precisely envelop the actual wound. The final segmented and cut wound is shown in Figure 19d.

The wound perimeter was measured as 44.07 mm, the area as 150.97 mm^2^, and the volume as 414.08 mm^3^.

#### 5.1.5. Larger Venous Ulcer

This wound is a venous ulcer of substantial size. The wound is rather flat, therefore no volume is measured. Due to some protrusions on the surface, two recordings were needed to reconstruct the surface shown on Figure 20a. Figure 20b shows the optimized mask projection that envelops a much larger area than the actual wound, but the ACM manages to converge on the actual wound outline despite the dynamic nature of that part of the surface, as can be seen in Figure 20c. Figure 20d shows the segmented and cut wound.

The wound perimeter was measured as 130.18 mm, while the area was 1324.47 mm^2^.

#### 5.1.6. Smaller Venous Ulcer

This is also a venous ulcer of relatively small proportions and, similar to the previously discussed venous ulcer, this one is also very flat and therefore no volume is measured. Only one recording was needed to generate the reconstruction shown in Figure 21a. The initial contouring made by the mask projection is shown in Figure 21b, and as can be seen, the masked area does not correctly overlap the wound area due to inaccuracy in the 2D wound segmentation and errors in camera calibration. Figure 21c shows that the ACM has somewhat correctly contoured the wound, with the only big error being that the top of the contour converged on another wound (a lipodermatosclerosis wound) that is in close proximity to this venous ulcer model. The other wound has a much bigger ridge, which results in a much larger curvature that is therefore “stealing” the contour. If this wound was not located that closely, the ACM would converge on the wound under consideration. The problem might also be fixed by using a larger α value of the ACM energy term, but in this research, we the used same configuration for all experiments. The final segmented and cut wound is shown in Figure 21d.

The wound perimeter was measured as 102.44 mm, while the area was 698.56 mm^2^.

## 6. Conclusions

Automated and precise wound measurements will improve the quality of the tools available to physicians to track the healing of individual patients and wounds. This would also facilitate better suited prescription and application of therapies, which would increase the living standards of patients as well as increase the healing rate of wounds, resulting in a reduction of costs in the medical system.

The segmentation system presented in this research is built upon on an automated, robot-driven acquisition system that outputs precise 3D reconstructions of chronic wounds. The acquisition system is fully automated and does not require any user input other than turning the robot in the general direction of the patient. The research presented here continues this philosophy of automation such that it does not require any user input in order for it to output accurate wound segmentation and geometric measurements. The data requested by the segmentation algorithm is the reconstructed 3D point cloud model of the wound as well as recordings and camera poses used to create the reconstruction. The recordings are used to create binary masks using the MobileNetV2 classifier and GrabCut, which label a general area of the wound on 2D images. These binary masks are then projected onto the 3D point cloud wound model, and then each point in the point cloud selects which projection to use based on the computed score, as well as the occlusion detection. Such a masked point cloud is used to generate an initial contour of the wound, which is then further refined by ACM on the meshed 3D surface of the reconstructed wound. The wound is then segmented, cut out, and the geometric measures of perimeter, area, and volume are calculated. 

Comparing the obtained measurement results with the authors’ previous work in [4], the average error rate was 2.82% for circumference, 5.72% for area, and 6.49% for volume measurement, compared to 4.39% for circumference, 6.86% for area, and 23.82% for volume.

The results presented clearly show that the segmentation and measurements are accurate, but further improvements must be made, especially regarding the reduction of errors caused by the calibration of cameras as well as 2D segmentation. Moreover, 3D ACM implementation could also be further improved by increasing the robustness to curvature noise as well as increased flexibility to better outline the arbitrary shapes that wounds can have.

## Figures and Tables

**Figure 1 sensors-23-03298-f001:**
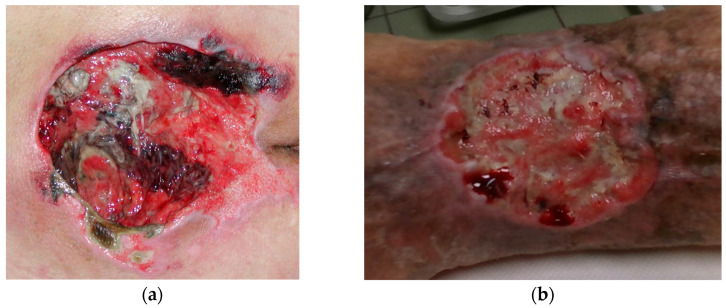
Wound located on: (**a**) lower back region; (**b**) leg region.

**Figure 2 sensors-23-03298-f002:**
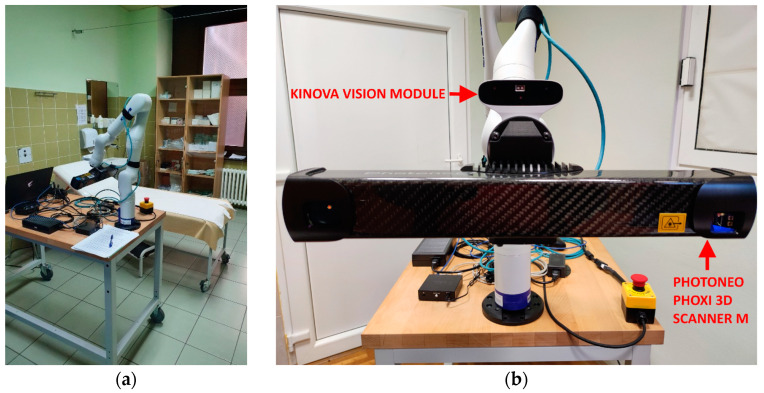
Configuration of the acquisition system: (**a**) robotic recording system in a hospital setting; (**b**) cameras used in the recording system.

**Figure 3 sensors-23-03298-f003:**
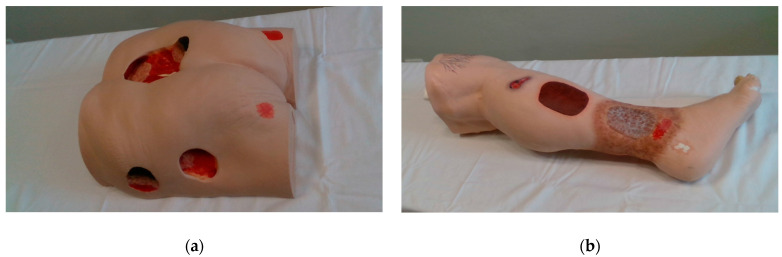
Medical models used in this research: (**a**) Seymour II wound care model; (**b**) Vinnie venous insufficiency leg model.

**Figure 4 sensors-23-03298-f004:**
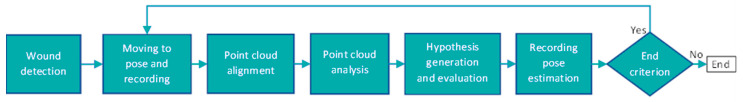
High-level overview of the wound detection and 3D reconstruction system.

**Figure 5 sensors-23-03298-f005:**
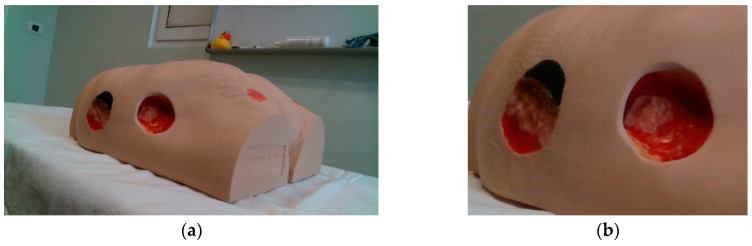
Sample image: (**a**) whole image; (**b**) enlarged section of the original image.

**Figure 6 sensors-23-03298-f006:**
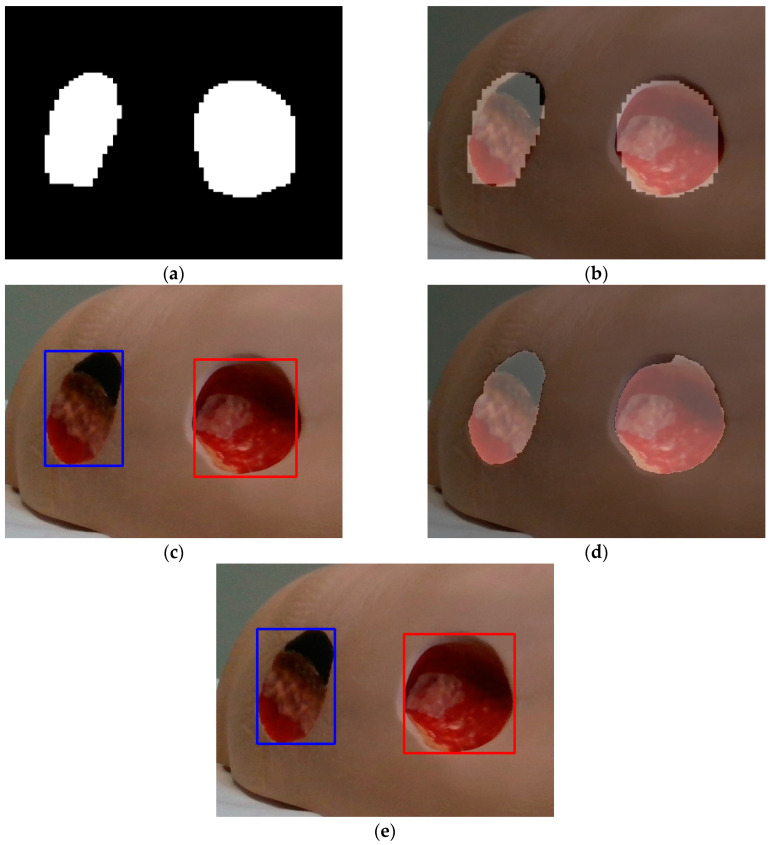
Enlarged section of the original test image under consideration: (**a**) output binary image obtained as a result of the method proposed by Wang et al. [10]; (**b**) wound pixels marked on the original image; (**c**) corresponding region of interests (ROIs) of image (**b**) where blue and red denotes different wounds; (**d**) wound pixels marked on the original image after additional post-processing using GrabCut; (**e**) corresponding ROIs of image (**d**) where blue and red denotes different wounds.

**Figure 7 sensors-23-03298-f007:**
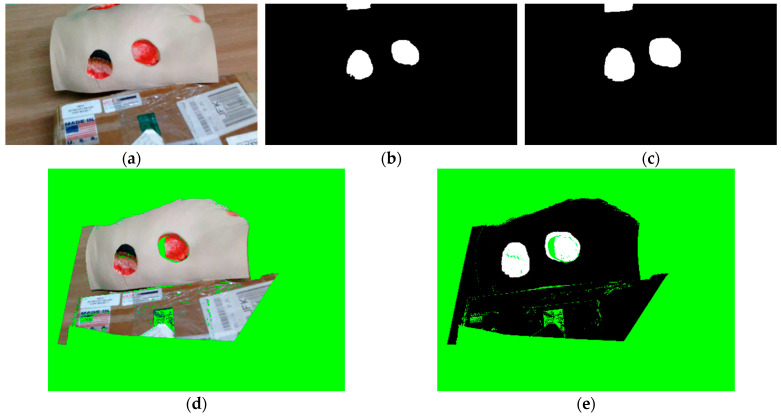
Example of registering binary masks to PhoXi depth images: (**a**) input RGB image made by Kinova vision module; (**b**) binary mask made during the 2D segmentation stage; (**c**) dilated binary mask; (**d**) registered RGB image—green color denotes pixels with no RGB value; (**e**) registered binary mask.

**Figure 8 sensors-23-03298-f008:**
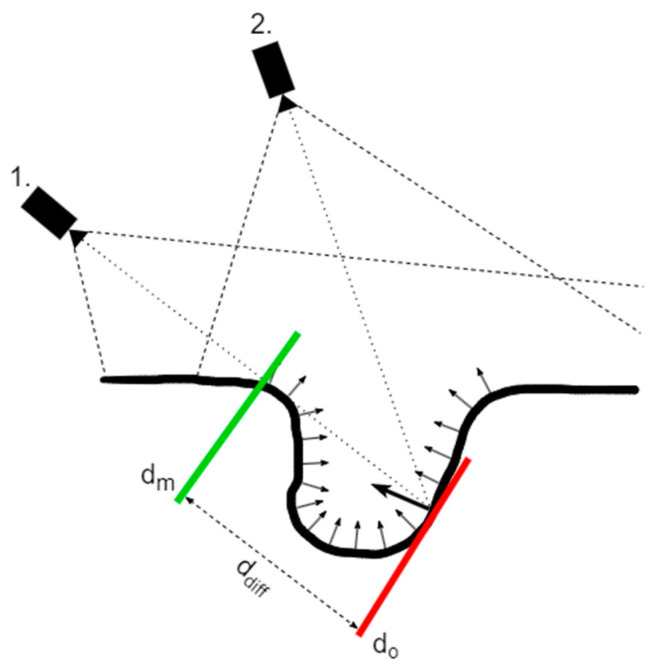
Occlusion detection when reprojecting surface 3D points for two positions (1., 2.), arrows represent surface normals while color bars represents locations for measured depth (green) and observed depth (red).

**Figure 9 sensors-23-03298-f009:**
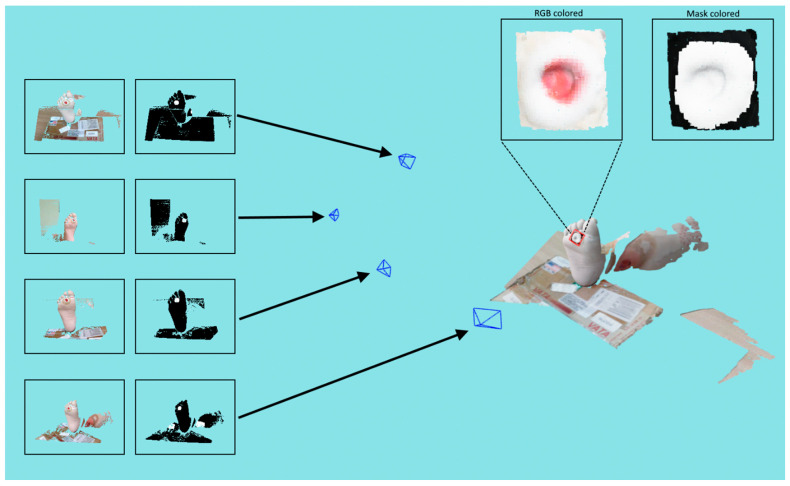
Example of optimized 3D point cloud mask labeling, where arrows point to the corresponding locations of the recordings on the left side of the figure.

**Figure 10 sensors-23-03298-f010:**
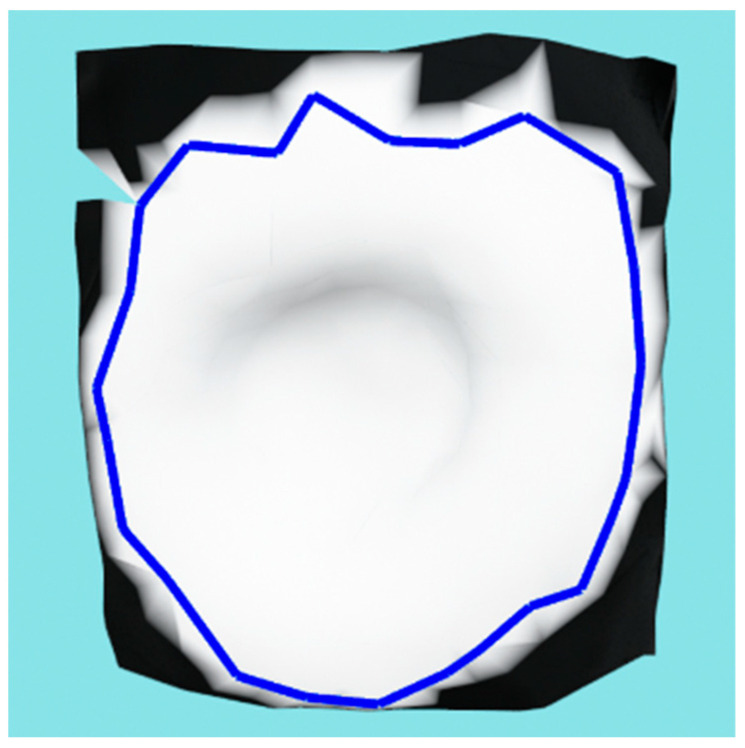
Initial wound contour shown in blue color on the meshed wound surface with binary mask texture.

**Figure 11 sensors-23-03298-f011:**
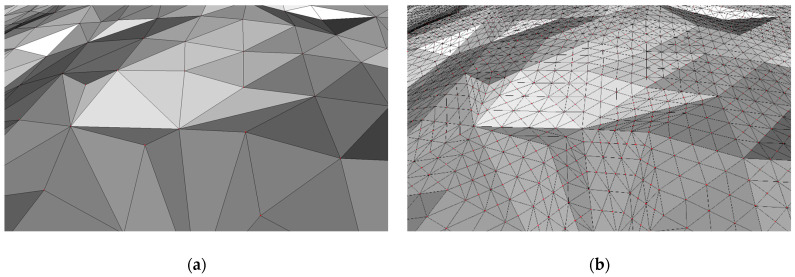
Example of using Midpoint algorithm for mesh subdivision: (**a**) original mesh; (**b**) subdivided mesh.

**Figure 12 sensors-23-03298-f012:**
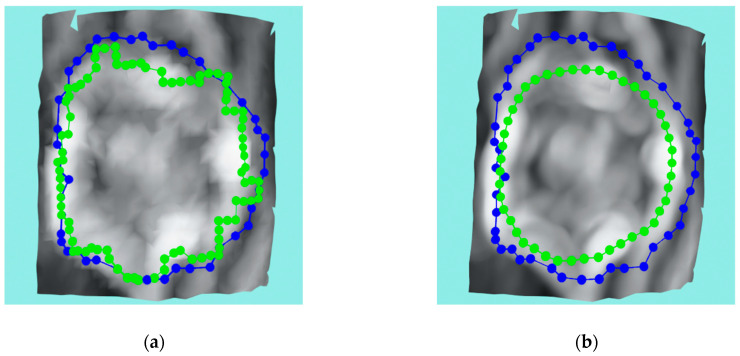
Example of improvement of ACM performance using mesh subdivision where blue represents initial contour while green is the final contour: (**a**) no subdivision used; (**b**) Midpoint subdivision used.

**Figure 13 sensors-23-03298-f013:**
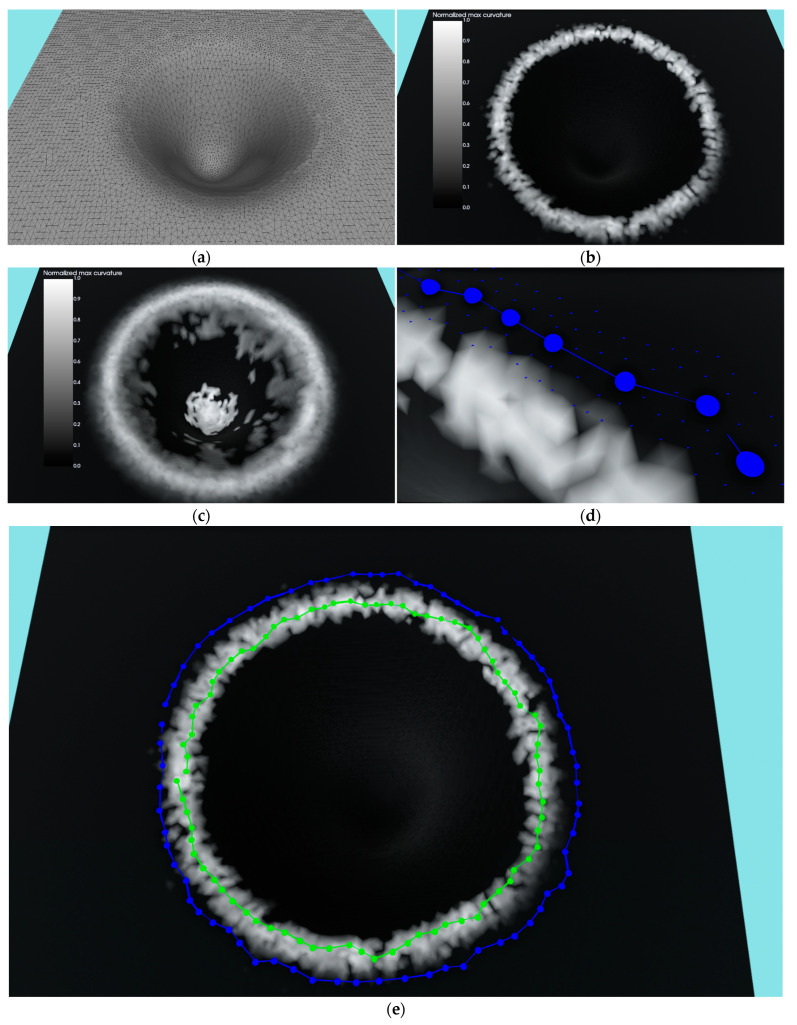
Synthetic example of ACM application: (**a**) input mesh of a hole created in a program; (**b**) curvature texture when using smaller neighborhood radius for calculating PCA; (**c**) curvature texture when using larger neighborhood radius for calculating PCA; (**d**) initial contour with nodes and their two-node neighborhoods; (**e**) initial and final contour (after ACM) with nodes colored in blue and green, respectively.

**Figure 14 sensors-23-03298-f014:**
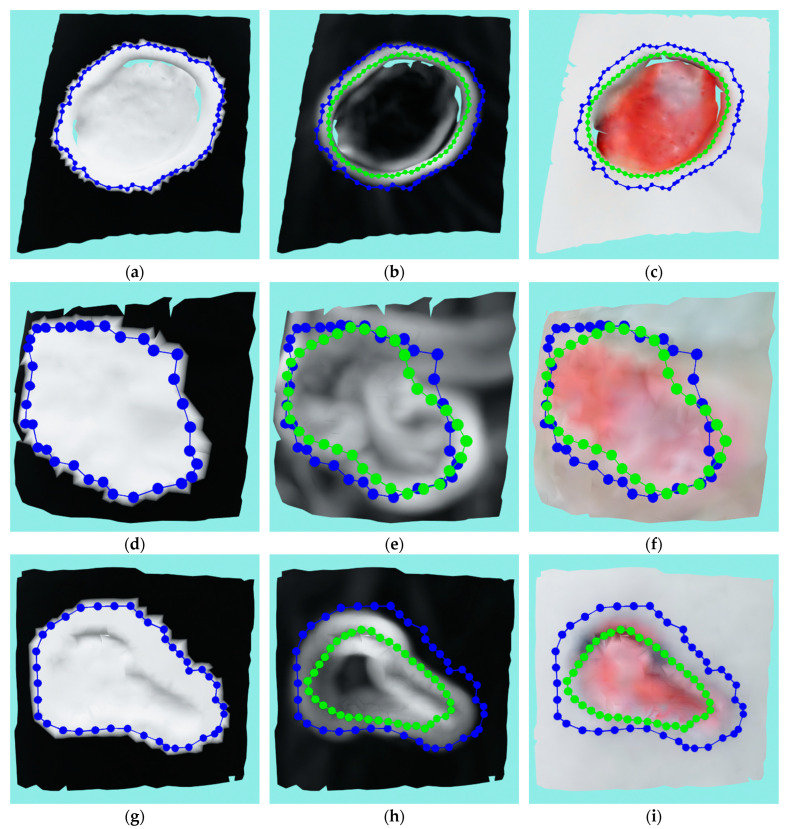
Real example of the ACM application where blue color represent initial contours while green color represent final contours: (**a**) masked mesh and initial contour on the successful run; (**b**) mesh with curvature texture and both initial contour and final contour on the successful run; (**c**) mesh with RGB texture and both initial contour and final contour on the successful run; (**d**) masked mesh and initial contour on the semi-successful run; (**e**) mesh with curvature texture and both initial contour and final contour on the semi-successful run; (**f**) mesh with RGB texture and both initial contour and final contour on the semi-successful run; (**g**) masked mesh and initial contour on the unsuccessful run; (**h**) mesh with curvature texture and both initial contour and final contour on the unsuccessful run; (**i**) mesh with RGB texture and both initial contour and final contour on the unsuccessful run.

**Figure 15 sensors-23-03298-f015:**
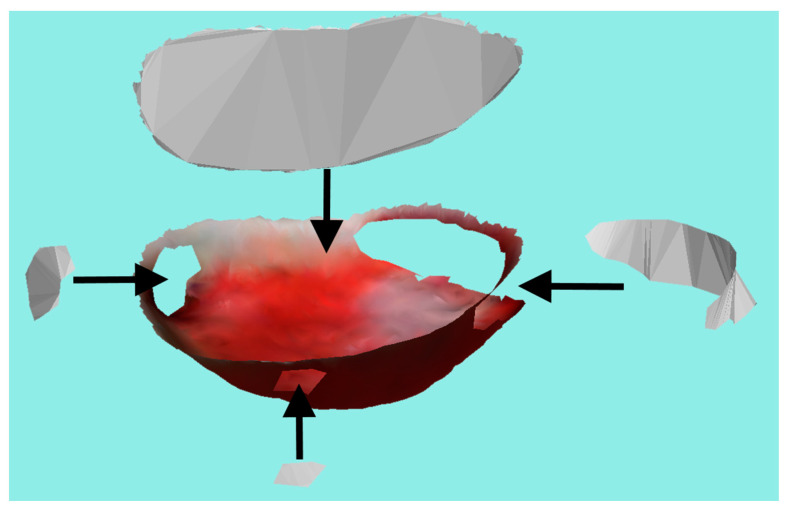
Final wound surface and created surfaces used for hole filling where arrows point to the actual location of the created surfaces.

**Figure 16 sensors-23-03298-f016:**
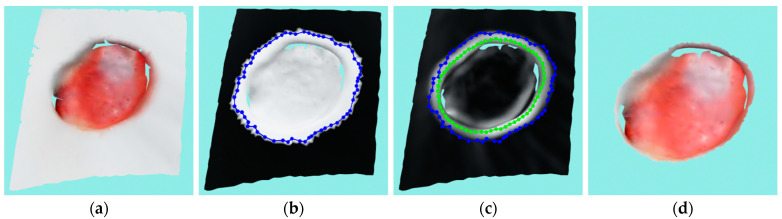
Case study for stage 3 pressure ulcer: (**a**) input wound mesh; (**b**) masked wound mesh with initial contour; (**c**) wound mesh with curvature texture and initial and final contours shown in blue and green colors, respectively; (**d**) final cut wound mesh model.

**Figure 17 sensors-23-03298-f017:**
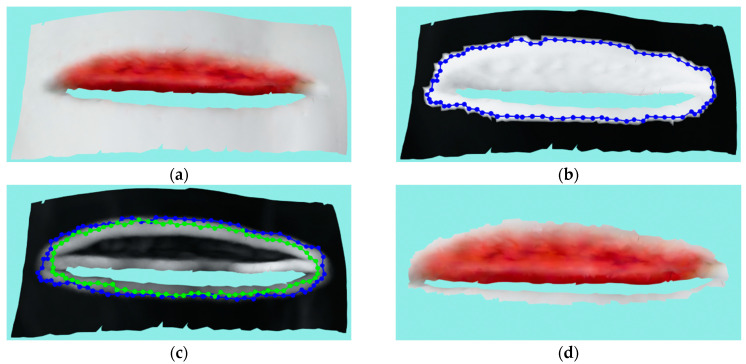
Case study for 5 ½ inch long dehisced surgical wound: (**a**) input wound mesh; (**b**) masked wound mesh with initial contour; (**c**) wound mesh with curvature texture and initial and final contours shown in blue and green colors, respectively; (**d**) final cut wound mesh model.

**Figure 18 sensors-23-03298-f018:**
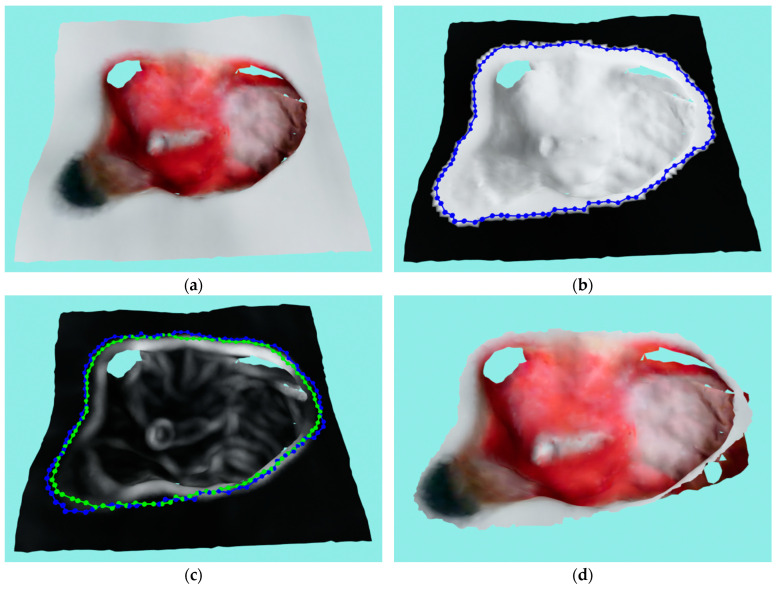
Case study for stage 4 pressure ulcer: (a) input wound mesh; (**b**) masked wound mesh with initial contour; (**c**) wound mesh with curvature texture and initial and final contours shown in blue and green colors, respectively; (**d**) final cut wound mesh model.

**Figure 19 sensors-23-03298-f019:**
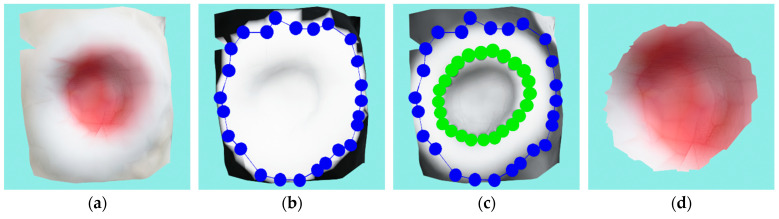
Case study for a neuropathic ulcer: (**a**) input wound mesh; (**b**) masked wound mesh with initial contour; (**c**) wound mesh with curvature texture and initial and final contours shown in blue and green colors, respectively; (**d**) final cut wound mesh model.

**Figure 20 sensors-23-03298-f020:**
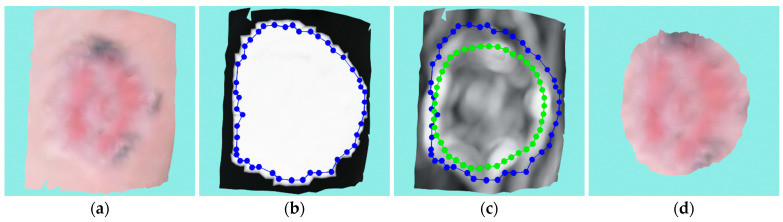
Case study for a larger venous ulcer: (**a**) input wound mesh; (**b**) masked wound mesh with initial contour; (**c**) wound mesh with curvature texture and initial and final contours shown in blue and green colors, respectively; (**d**) final cut wound mesh model.

**Figure 21 sensors-23-03298-f021:**
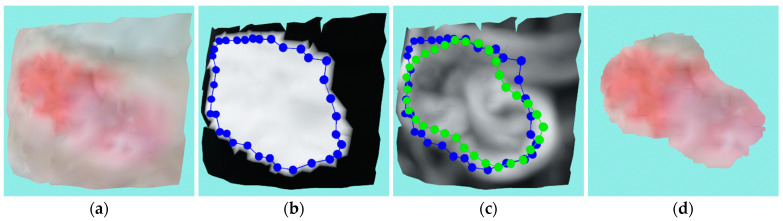
Case study for a smaller venous ulcer: (**a**) input wound mesh; (**b**) masked wound mesh with initial contour; (**c**) wound mesh with curvature texture and initial and final contours shown in blue and green colors, respectively; (**d**) final cut wound mesh model.

**Table 1 sensors-23-03298-t001:** Ground truth for the measurements of Seymour II wound care model.

Wound Type	Perimeter (mm)	Area (mm^2^)	Volume (mm^3^)
Stage 4 pressure ulcer	334.9	7117.6	100,821.2
Stage 3 pressure ulcer	175.7	2369.8	21,542.8
5 ½ inch long dehisced surgical wound	277.7	2400.0	28,090.9

## Data Availability

The data presented in this study are available on request from the corresponding author.

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
