# Peer review of "2D/3D Wound Segmentation and Measurement Based on a Robot-Driven Reconstruction System"

_sensors, 2023, doi:10.3390/s23063298_

Round 1

Reviewer 1 Report

Dear editor,

 The manuscript describes an automatic wound segmentation based on a wound recording system built upon a 7 DoF robot arm with an attached RGB-D camera and high precision 3D scanner. The developed system represents  a novel combination of 2D and 3D segmentation, where the 2D segmentation is based on the MobileNetV2 classifier and the 3D component is based on the active contour model, which works on the 3D mesh to further refine the wound contour. The end output is the 3D model of only the wound  surface without the surrounding healthy skin and geometric parameters in the form of perimeter, area and volume. This manuscript is very innovative, but I cannot recommend it to be published in this journal in the current format,The main reasons include the following: the paper lacks accurate qualitative analysis of the nature of the wound, the reference index for reconstruction is vague, and the prediction of the method is poor. Therefore, the author needs to make corresponding improvements in these aspects and should give corresponding indicators, Its detailed explanation provides the reliability and practical basis of the indicators.

Author Response

Reviewer 1

The manuscript describes an automatic wound segmentation based on a wound recording system built upon a 7 DoF robot arm with an attached RGB-D camera and high precision 3D scanner. The developed system represents  a novel combination of 2D and 3D segmentation, where the 2D segmentation is based on the MobileNetV2 classifier and the 3D component is based on the active contour model, which works on the 3D mesh to further refine the wound contour. The end output is the 3D model of only the wound  surface without the surrounding healthy skin and geometric parameters in the form of perimeter, area and volume. This manuscript is very innovative, but I cannot recommend it to be published in this journal in the current format The main reasons include the following: the paper lacks accurate qualitative analysis of the nature of the wound, the reference index for reconstruction is vague, and the prediction of the method is poor. Therefore, the author needs to make corresponding improvements in these aspects and should give corresponding indicators, Its detailed explanation provides the reliability and practical basis of the indicators.

Response:

“…the paper lacks accurate qualitative analysis of the nature of the wound,…”

The goal of this manuscript is to research and develop a segmentation algorithm that will allow qualitative analysis of the wound in later steps. Therefore, only geometric measurement is currently available when separating wound area from healthy skin. Qualitative wound analysis, including tissue type analysis of the wound surface, is our next step, and hopefully within a year we will have a working subsystem within our robotic platform that will allow such analysis on reconstructed 3D models.

“…the reference index for reconstruction is vague,…”

We are not quite sure what is meant by "reference index for reconstruction," so we can only assume that the reviewer is referring to ground truth data/measurements. The ground truth data for the research using the VATA Seymour II wound care model was determined by a company that specializes in accurate 3D reconstruction of machine parts. They used GOM ATOS technology and matting coating to achieve extremely accurate reconstruction of the shiny/glossy wound surface. The use of matting coatings on actual wound surfaces is not possible due to the very high risk of contamination. This means that the ground truth created in this case will always have a more accurate 3D reconstruction than projection-based system 3D scanners such as we use in this research due to the reflective properties of the wound surface.

If by "reference index for reconstruction" the reviewer instead means the shortened explanation of the operation of the 3D reconstruction system in Section 4.1, it is because it was intended only as a brief summary of the operation of the reconstruction system. The full description, as well as the ground truth accuracy, can be found in the freely available full text: https://www.mdpi.com/1424-8220/21/24/8308

“…the prediction of the method is poor.”

As mentioned earlier, ground truth was established by applying a matting spray coating (e.g., Helling Entwickler No.3 or Helling Nord-test U89) to the medical model, which greatly improves 3D scanning with projection-based 3D scanners by preventing reflections. The application of such a spray on living tissue, especially wounds, would lead to very unhealthy effects. Therefore, the accuracy of 3D scanners on most wounds is lower because of the shiny parts of the wound surface. Regardless, the accuracy of the reconstruction system is quite high compared to ground truth, with the calculation of the distance between the reconstructed point cloud and the ground truth mesh yielding statistical data with a mean error of 0.14 mm and a standard deviation of 0.12 mm, meaning that about 64% of the points had an error of less than 0.15 mm and almost 99% of the points had an error of less than 0.5 mm. Details on the precision of the 3D reconstruction module can be found in:

Filko, D.; Marijanović, D.; Nyarko, E.K.. Automatic robot-driven 3D reconstruction system for chronic wounds. Sensors 695, 2021, 21, 8308. https://doi.org/10.3390/s21248308

“…and should give corresponding indicators, Its detailed explanation provides the reliability and practical basis of the indicators”

We do not understand this statement/requirement made by the reviewer, hence we do not know what is required of us.

Reviewer 2 Report

Dear authors, 

Your manuscript is well written, but is very similar with your previous articles. 

This article does not bring anything new for literature.  

Author Response

Reviewer 2

Dear authors,

Your manuscript is well written, but is very similar with your previous articles.

This article does not bring anything new for literature. 

Response:

“… but is very similar with your previous articles.”

Research and development of more accurate and reliable wound analysis systems is an ongoing process. Therefore, there must be some similarities to our previous research, simply by the fact that nearly 15 years of research has been conducted in the same field by us. The field of wound analysis is constantly changing and evolving with the advent of new technologies that can be applied to this problem. Our earlier work, which is most comparable to this research, can be found here:

Filko, D.; Cupec, R.; Nyarko, E.K.. Wound measurement using an RGB-D camera. Mach. Vis. Appl. 2018, 29, 633-654.

The referenced work also focuses on 3D reconstruction, segmentation, and measurement of chronic wounds, but uses very different technologies than the current research. Previous research used handheld RGB-D cameras, which were significantly cheaper but had significant drawbacks in terms of depth accuracy and the influence of surface features and lighting conditions. Because they were handheld cameras, the accuracy of the reconstruction was also affected by the experience of the operator. In the current research, a sophisticated 7DoF robotic arm is used with an industrial high-precision 3D scanner attached to the end effector to enable a fully automated and accurate 3D reconstruction process. The research presented in this paper is the continuation of the development of the automatic wound assessment system with the implementation of the novel wound segmentation process.
In summary, it is a logical continuation of our previous work and research in general on this topic. A statement to this effect has been added on page 2 of the paper

“This article does not bring anything new for literature.” 

The research presented in this paper consists of a novel combination of 2D CNN and 3D geometry-based segmentation. To our knowledge, no one in this field has used this combination of technologies to achieve accurate segmentation of chronic wounds on 3D reconstructions. As mentioned earlier, this research represents the logical extension of previously published work. When we directly compare the measurement results with previous work, we can find an average error rate of 2.82% for circumference, 5.72% for area, and 6.49% for volume measurement for the current research, while our previous research achieved an average error rate of 4.39% for circumference, 6.86% for area, and 23.82% for volume. A statement to this effect has been added in the conclusion of the paper. Furthermore, this can be used for discussion when comparing the results of the hand-held system with a fully automated robotic system, which we believe is relevant in the literature.

Reviewer 3 Report

Review comments for 2D/3D wound segmentation and measurement based on a robot driven reconstruction system

We thank author for investigating a novel segmentation algorithm by using both 2D and 3D procedures to develop a 3D wound measurement model for better wound size/depth detection. This research paper is well-written and informative. Every step and description is organized and clear. Some minor concerns need to be addressed first before publication.

Is well written, step by step description is clear and informative.

1.     It would be beneficial to add scale bars to the digital analyzed images.

2.     In section 5, author listed several critical wounds, could author also include a wounds that from the foot volar?

Author Response

Reviewer 3

Review comments for 2D/3D wound segmentation and measurement based on a robot driven reconstruction system

We thank author for investigating a novel segmentation algorithm by using both 2D and 3D procedures to develop a 3D wound measurement model for better wound size/depth detection. This research paper is well-written and informative. Every step and description is organized and clear. Some minor concerns need to be addressed first before publication.

Is well written, step by step description is clear and informative.

  1. It would be beneficial to add scale bars to the digital analyzed images.
  2. In section 5, author listed several critical wounds, could author also include a wounds that from the foot volar?

Response:

“1. It would be beneficial to add scale bars to the digital analyzed images.”

As requested by the reviewers, we have added scale bars to Figures 13b and 13c showing the scale of normalized maximum curvature for the general examples of ACM use. We did not add scale bars to the other figures showing curvature because it would not be useful due to the small size of some of the figures, particularly in Section 5, and would also distract from viewing the other data on the figures.

“2. In section 5, author listed several critical wounds, could author also include a wounds that from the foot volar?”

From the Vinnie venous insufficiency leg model, we included 2 venous ulcers (minor and major) and 1 neuropathic ulcer as case studies; we did not include lipodermatosclerosis and stasis dermatitis because they belong to inflammatory skin diseases rather than wounds and are therefore not currently part of the research project conducted. More details about the available wounds of the medical model can be found on the manufacturer's website: https://vatainc.com/product/vinnie-venous-insufficiency-leg-model/

Round 2

Reviewer 1 Report

OK,I recommend that the manuscript be published in this journal in the current format, but the format of the manuscript should be reorganized.

Reviewer 2 Report

Dear authors, 

Your answers clarified my doubts.

Congrats.